

# Autonomous marine hyperspectral radiometers for determining solar irradiances and aerosol optical properties

John Wood[1], Tim J. Smyth[2] and Victor Estellés[3]

[1]Peak Design Ltd, Sunnybank House, Wensley Rd, Winster, Derbys. DE4 2DH UK
5   [2]Plymouth Marine Laboratory, Prospect Place, Plymouth, Devon, PL1 3DH, UK
[3]Dept. Física de la Terra i Termodinàmica, Universitat de València, Burjassot, 46100, Spain

*Correspondence to*: Tim Smyth (tjsm@pml.ac.uk)





**Abstract.** We have developed two hyperspectral radiometer systems which require no moving parts, shade rings or motorised tracking making them ideally suited for autonomous use in the inhospitable remote marine environment. Both systems are able to measure the direct and diffuse hyperspectral irradiance fields in the wavelength range 350 – 1050nm at 6nm (Spectrometer 1) or 3.5nm (Spectrometer 2) resolution. Marine field-trials along a 100° transect (between 50°N and 50°S) of the Atlantic Ocean resulted in close agreement with existing commercially available instruments in measuring: (1) photosynthetically available radiation (PAR) with both spectrometers giving regression slopes close to unity (Spectrometer 1: 0.960; Spectrometer 2: 1.006) and $R^2$ ~0.96; (2) irradiant energy, with $R^2$~0.98 and a regression slope of 0.75 which can be accounted for by the difference in wavelength integration range and; (3) hyperspectral irradiance where the agreement on average was between 2 – 5%. Two long duration land based field campaigns of up to 18 months allowed both spectrometers to be well calibrated. This was also invaluable for empirically correcting for the wider field-of-view (FOV) of the spectrometers in comparison with the current generation of sun photometers (~7.5° compared with ~1°). The need for this correction was also confirmed and independently quantified by atmospheric radiative transfer modelling and found to be a function of aerosol optical depth (AOD) and solar zenith angle. Once Spectrometer 2 was well calibrated and the FOV effect corrected for, the RMSE in retrievals of AOD when compared with a CIMEL sun photometer were reduced to ~0.02 – 0.03 with $R^2 > 0.95$ at wavelengths 440, 500, 670 and 870nm. Corrections for the FOV as well as ship motion were applied to the data from the marine field trials. This resulted in $AOD_{500nm}$ ranging between 0.05 in the clear background marine aerosol regions to ~0.5 within the Saharan dust plume. The RMSE between the handheld Microtops sun photometer and Spectrometer 2 was between 0.047 – 0.057 with $R^2 > 0.94$.

**Copyright statement**

**1. Introduction**

Tiny particles within the atmosphere, collectively known as aerosols, play a key role in the functioning of the Earth System as a whole. However, a great deal of uncertainty remains concerning precise and quantifiable mechanisms within that



system. These mechanistic uncertainties generally fall into the broad categories of: aerosol sources and subsequent sinks; aerosol transformational mechanisms (e.g. from aerosol to cloud condensation nuclei) and; aerosol types. Aerosol type is determined by its source region and in turn this determines its singular and integrated physical attributes. For example soot particles produced by natural or anthropogenic combustion are generally small in size, have a low single scattering albedo

and are subsequently highly absorbing in the optical region of the electromagnetic spectrum. Near the source regions these aerosols are small (<1µm) and high in number. In contrast, aerosols produced in the marine environment by breaking waves, wind driven spume and bubble bursting are generally large (up to 10 µm) but relatively low in number. They have a high single scattering albedo (> 0.95) and hence absorb a relatively small proportion of incoming solar radiation. Just from these two simple examples it can be seen that aerosol type will have a large bearing on the local, regional and global radiative

balance, and why a large uncertainty still exists our understanding of precisely how aerosols impact Earth's climate as a whole.

During the past twenty years advances have been made in measuring aerosol optical properties over the terrestrial parts of the globe. These include the AERONET (Holben et al., 1998), SKYNET (Takamura and Nakajima, 2004) and ESR (Campanelli et al., 2012) networks which employ sun photometric techniques to determine multi-spectral aerosol optical

depth and their physical characteristics (refractive index, single scattering albedo, size distribution) by radiative inversion schemes (Nakajima et al., 1996;Dubovik and King, 2000). Although these networks are particularly densely populated in North America, Eastern Asia and Europe, there is very little or non-existent coverage over the vast expanses of the global ocean. This is due in part to the difficulty in using a moving platform such as a ship to get an accurate fix on the position of the sun using a small, columnar field of view (typically ~ 1°). Recent expansion by the AERONET network to cover the

remote global ocean (Smirnov et al., 2009) has ameliorated this situation somewhat, however the instruments typically used at sea rely upon handheld sun photometers, such as the Microtops (Morys et al., 2001), which by definition require a human operator. This generally limits the number of ships of opportunity which carry such devices to scientific research expeditions. Ideally an autonomous instrument needs to be developed which can potentially be deployed on any ship or platform to cover the considerable gaps, spatial and temporal, in the ocean aerosol observing network.



The solar radiation measurement can be split into three components: the global horizontal irradiance $I_G$, direct normal irradiance $I_N$ and diffuse horizontal irradiance $I_D$. The current state-of-the-art measurement (McArthur, 2005), uses a pyrheliometer on a solar tracker to measure $I_N$, and pyranometers (one shaded by a tracker-mounted ball) to measure $I_G$ and $I_D$ respectively. However, such an instrument combination requires an initial high capital outlay and requires frequent and

complex onsite maintenance. Other options include a pyranometer for $I_G$ and pyranometer with shade ring for $I_D$ with $I_N$ being calculated from these two components. The shade ring requires regular adjustment and a correction applied for the shaded part of the diffuse sky. Pertinent to this work, rotating shadowband radiometers which use a silicon photodiode detector and a motorised rotating shading ring to measure both $I_G$ and $I_D$, have been used in the marine environment to determine aerosol optical properties (Reynolds et al., 2001). Assuming a clear sky, the aerosol optical depth, $\tau_a$, can then be

calculated.

In this paper we describe a similar concept, but with the following important differences in construction: (1) use of a unique etched shadow design (Badosa et al., 2014), to remove the need for moving parts for splitting the irradiance into the global and diffuse components; (2) use of hyperspectral radiometers to give finer spectral detail and hence aerosol optical characterisation. Difference (1) is particularly important in the harsh marine environment over prolonged periods of

autonomous operation as salt spray can quickly seize moving parts as can freezing temperatures. We describe methods for accurate calibration of the instruments; demonstrate their operational robustness on an Atlantic Meridional Transect cruise (AMT24: 22 September – 01 November 2014) between the UK and the Falkland Islands; carry out an intercomparison between existing field-based instruments and; highlight operational issues and propose solutions.

The structure of this paper is as follows. A methods section (Section 2) describing the theoretical basis (Section

2.1) and technological implementation (Section 2.2) of our approach together with the calibration (Section 2.3) and aerosol optical depth intercomparison (Section 2.4). The results section (Section 3) focusses in the main on the AMT24 cruise and the associated issues encountered when developing and improving instrumentation in the field. This included: correcting the measurements for orientation (Section 3.1) and intercomparison with co-located established radiometric instrumentation (Section 3.2). Following the AMT24 cruise it was found that the data were improved by using land-based intercomparison

studies pre- and post- cruise (Section 3.3) and allowed us to determine the theoretical and empirical basis for correcting for



field of view differences (Section 3.4). Finally all these findings and strategies are brought together (Section 3.5) to process

the hyperspectral radiometer system data to derive aerosol optical depth.

## 2. Method

### 2.1 Theory

Devices to measure irradiance typically report raw values as voltages (V), thus:

$$V_G(\lambda) = V_H(\lambda) + V_D(\lambda) \qquad (1)$$

where $\lambda$ is wavelength, G is global, H is horizontal direct, and D is diffuse. See Table 1 for a glossary of symbols and

definitions. The volts direct onto the horizontal plane, $V_H(\lambda)$, are normalised by the solar zenith angle ($\theta_s$) using:

$$V_N(\lambda) = V_H(\lambda) \sec \theta_s \qquad (2)$$

The instrument can be calibrated against known standard instruments in the laboratory or in the field. It is also necessary to

carry out a Langley calibration (Adler-Golden and Slusser, 2007) of the instrument during clear and stable atmospheric

conditions over the course of a day using Beer's Law to obtain the top of atmosphere voltage, $V_T(\lambda)$. This can be

represented as:

$$V_N(\lambda) = V_T(\lambda) \exp(-\tau(\lambda)m) \qquad (3)$$

where $\tau(\lambda)$ is the optical depth and m is the atmospheric air-mass defined as:

$$m = \frac{1}{\cos \theta_s + a(b - \theta_s)^{-c}} \qquad (4)$$

In Equation 4 the constants a, b, c are set to 0.50572, 96.07995 and 1.6364 respectively (Kasten and Young, 1989). To

account for the elliptical nature of Earth's orbit the following expression is used:

$$V_{0T}(\lambda) = V_T(\lambda)r^2 \qquad (5)$$

where:

$$r = (1 - \varepsilon \cos(a[J - 4])) \qquad (6)$$





J being the serial day of the year, $\varepsilon$ is the eccentricity of the orbit (0.01673) and $a=2\pi/365.25$ Expanding Equation 3 into the component parts of the optical depth, Rayleigh (R), aerosol (a) and atmospheric trace gases (g) results in:

$$V_N(\lambda) = \frac{V_{0T}(\lambda)}{r^2} \exp(-(\tau_R(\lambda) + \tau_a(\lambda) + \tau_g(\lambda))m) \qquad (7)$$

The trace gas component (such as ozone, nitrous oxide, water vapour) can be derived from measurements or distribution climatologies in conjunction with models such as SMARTS2 (Gueymard, 2001); the Rayleigh component can be calculated from (Reynolds et al., 2001):

$$\tau_R(\lambda) = \frac{P}{P_0}(a_1\lambda^4 + a_2\lambda^2 + a_3 + a_4\lambda^{-2})^{-1} \qquad (8)$$

where P is the atmospheric pressure (mb), $P_0 = 1013.25$ mb; $a_1 = 117.2594$; $a_2 = -1.3215$; $a_3 = 0.00032073$ and; $a_4 = -0.000076842$. Rearranging equation (7) allows the aerosol optical depth to be calculated for each individual optical wavelength. The trace gas components are not corrected for in this study.

## 2.2 Technological implementation

### 2.2.1 SPN1 Radiometer

The SPN1 (Wood, 1999) is a broadband radiometer without moving parts, shade rings or motorised tracking that measures $I_G$ and $I_D$ broadband short-wave irradiance (from 400 to 2700 nm) expressed in $Wm^{-2}$. The SPN1 was designed with seven thermopiles: six sensors placed on a hexagonal grid, one sensor at the centre, under a complex static shading mask (see Figure 1), in such a way to ensure that, at any time, for any location: (1) at least one sensor is always exposed to the full solar beam; (2) at least one sensor is always completely shaded and; (3) the solid angle of the shading mask is equal to $\pi$ thus corresponding to half of the hemispherical solid angle.

Under the assumption of isotropic diffuse sky radiance, the third property related to the shading mask implies that all sensors receive equal amounts (50%) of diffuse irradiance from the rest of the sky hemisphere. It can therefore be seen that at any instant, the minimum signal ($I_{min}$) measured among the seven sensors is the shaded sensor, which measures half





the $I_D$, and the maximum signal ($I_{max}$) from among the seven sensors is fully exposed to the solar beam, and therefore

measures the $I_H$ plus half the $I_D$. From this the following relationships can be formed:

$$I_D = 2I_{min} \qquad (9)$$

$$I_H = (I_{max} - I_{min}) \qquad (10)$$

$$I_G = I_H + I_D = I_{max} + I_{min} \qquad (11)$$

By calculating the (relative) solar zenith angle ($\theta_{rs}$) using the known time and geographical position, $I_N$ can be derived thus:

$$I_N = I_H sec\,(\theta_{rs}) \qquad (12)$$

For a detailed study of the performance of the SPN1 the reader is referred to Badosa et al. (2014).

**2.2.2 Spectrometers based on the SPN1**

10     In this study, the broadband detectors of the SPN1 have been replaced by spectrometers to give hyperspectral measurements

of $I_G(\lambda)$, $I_D(\lambda)$ and $I_N(\lambda)$ over the range 350nm – 1050nm. Light is collected from behind the diffuser elements of the SPN1

optical head, and routed to a spectrometer via an optical fibre. In order to evaluate the various trade-offs between cost, speed

of measurement, and consistency of measurement, prototypes of two different configurations were constructed (see Figure

2).

15     **2.2.3 Spectrometer system 1 – AS161**

In this configuration, the seven optical fibres were each routed directly to one of seven low-cost optical benches

manufactured by Avantes, and controlled by the Avantes AS161 control board. These optical benches had 128 pixel

detectors giving a pixel resolution of around 6nm across the range 350nm – 1050nm. The advantage of this configuration is

that all seven optical channels can be read in parallel in a short time (<1s), therefore removing many of the potential artefacts

20     due to making measurements on a moving platform. The main disadvantages are that: (1) a cheaper spectrometer is

required; (2) it is more difficult to maintain a close matching between spectrometer calibrations and; (3) the wavelengths

corresponding to each pixel are different for each measurement channel. N.B. Spectrometer 1 developed an electronics fault

towards the end of the AMT cruise, so a shorter period of comparison results is available than for Spectrometer 2.





### 2.2.4 Spectrometer system 2 – Zeiss

In this configuration, the seven optical fibres are taken via a fibre-optic multiplexer to a single Zeiss MMS1 spectrometer. This has a 256 pixel detector, giving a pixel resolution of around 3.5nm across the range 350nm – 1050nm. The advantage of this configuration is that the Zeiss is a very stable spectrometer over a wide range of temperatures, with a high sensitivity.

All seven optical channels are measured at the same sensitivity and set of wavelengths. The primary disadvantage of this configuration is that the seven optical channels are measured sequentially over a period of 20s in total. This means that irradiance variations due to cloud or movement occurring during the measurement period will compromise the accuracy of the overall measurement.

### 2.2.5 Control electronics and software

Both spectrometer systems are controlled by an embedded PC running Windows XP. There are also additional sensors to measure GPS position and time, atmospheric pressure, temperature, humidity within the enclosure. A heading, pitch and roll sensor was also included. The control software is responsible for reading the spectrometer values, sequencing the switch, and combining the values into calibrated measurements of $I_G(\lambda)$ and $I_D(\lambda)$, and recording these at the appropriate times (1-minute intervals), along with readings of the additional environmental sensors. The system is controlled via an Ethernet

connection. Each spectrometer system required a 12V power supply capable of 1A peak draw; all these components were packaged in a weatherproof enclosure.

### 2.2.6 Ship motion

Both spectrometer systems were fitted with a VectorNav VN100 inertial orientation sensor, containing three-axis sensors for each of linear acceleration, angular acceleration, and magnetic field. From these measurements, the sensor calculates values

of yaw (heading), pitch and roll. These measurements allowed the spectral measurements to be corrected for the tilt of the instruments away from the horizontal.

### 2.2.7 GPS position and time

Both spectrometers were fitted with GPS receivers, and the GPS time and position recorded throughout the cruise. The spectrometers were referenced to their own embedded PC clocks, and these showed drifts of several minutes over the



duration of the six week cruise. By referring all the readings to GPS time, it was possible to compare the various datasets using a consistent time base.

### 2.2.8 Data sampling and recording

The two spectrometer configurations required slightly different sampling and recording strategies.

*Spectrometer 1 – AS161.*

In this spectrometer, all seven measurement channels are read in parallel over a 500ms time span. To compensate for wave motion, a burst of ten readings is taken at one per second. The average of these ten readings is used for subsequent calculations, although the individual burst readings are available if necessary. A burst of readings is repeated every minute.

*Spectrometer 2 – Zeiss.*

In this spectrometer, the seven measurement channels are measured sequentially. Each channel takes approximately 3s, so a full measurement takes ~20s. At each channel reading, the SPN1 irradiance is also measured, along with orientation values from the VectorNav sensor. These values are used to improve the measurements by correcting for tilt during subsequent analysis.

### 2.2.9 Housing and mounting position

During the ship-based part of this study, the spectrometers were mounted on the top of the foremast of the British Antarctic Survey research ship *RRS James Clark Ross* on a dedicated instrument platform (Figure 3). Access was only possible via the ship's crane and hoist when in port at the beginning and end of the cruise, so once installed there was no further opportunity for modifications or maintenance. The spectrometers were both mounted in IP67 weatherproof enclosures, and fitted with desiccant packs. The heat generated by the electronics increased the internal temperature by around 10°C – 15°C above the

ambient, and this helped to keep the internal humidity to less than 30% during the cruise. An SPN1 radiometer was also mounted alongside the two spectrometers to give a broadband irradiance reference. The instruments were powered by a 12V power cable, and communications provided by an Ethernet cable, both routed up the mast. The performance of the spectrometers was monitored throughout the cruise, remotely from inside the ship, via the Ethernet connection. A Satlantic



Hyperspectral radiometer, Kipp & Zonen Photosynthetically Available Radiation (PAR) sensors and Kipp & Zonen pyranometers were also mounted on the instrument platform throughout the cruise.

### 2.2.10 Tilt correction strategy

On analysis of the orientation values after the cruise, the VectorNav yaw (heading) values showed significant drift compared to the yaw values calculated from both the GPS track, and the ship's heading record. This was due magnetic interference from the ship's ironwork, which had not been compensated for when the spectrometers were installed. However, the pitch and roll values could still be used in combination with yaw values either taken from the ship's data records after the cruise, or calculated from the GPS track values.

Long et al. (2010) demonstrated a method for correcting pyranometer measurements on an aircraft using SPN1 measurements. We have used a similar technique to correct both the SPN1 and Spectrometer 2 values in this study.

In correcting the Spectrometer 2 values, it is assumed that the diffuse part of the incident light is unaffected by tilt. The diffuse value is calculated from the minimum of the seven channels. This is subtracted from all the other channels to give the direct beam part of the reading on the instrument plane ($I_{Hmeas}$). The direct beam part is then corrected according to the known position of the sun, and the angle of incidence on the tilted instrument plane calculated from the orientation values.

$$I_{Hcorr} = \frac{I_{Hmeas}}{\cos\theta_{rs}}\cos\theta_s \qquad (13)$$

where

$$cos\theta_{rs} = \cos\theta_s\ \cos\alpha_{sf} +\ \sin\theta_s\sin\alpha_{sf}\cos(\phi_s - \beta_{sf}) \qquad (14)$$

See Table 1 for definition of the various angles. The seven channels are then recalculated from the $I_{min} + I_{Hcorr}$ and used to calculate the corrected $I_G$, $I_D$ and $I_N$ using equations 9-11. This correction is also applied to the SPN1 values.

There are two contributions to irradiance variation during the reading period – variations in the overall irradiance values (e.g. variable cloud cover, particularly obscuring the solar disc), and variations due to tilt of the ship. This correction strategy will correct for the ship's movement, but not variations in light levels during the reading period.





## 2.3 Calibration and traceability

There are two requirements for calibration of this spectrometer system. Firstly, the seven individual channels should have an identical response to incident light. Secondly the response should be matched to the absolute irradiance scale across the whole spectrum. To achieve this, the spectrometers were first calibrated using an integrating sphere to give a uniform

irradiance across all the sensors. The integrating sphere lamp was calibrated to an Ocean Optics LS-1 calibrated lamp to give an approximately correct overall calibration. Following this, the spectral calibration was adjusted using the Langley method on Mt Teide, Tenerife (2300m, near the base of the teleferico), to give a final absolute calibration.

After the AMT24 cruise, Spectrometer 2 (Zeiss) was co-located with a CIMEL sun photometer in Burjassot (Valencia, Spain: 39° 30.58′ N, 0° 25.08′W) for 18 months. Its calibration was further checked using the Langley method

during selected clear-sky periods, and also by a direct comparison with the CIMEL $I_N(\lambda)$ measurements at the specific CIMEL wavelengths. Figure 4 shows how these different methods compare, by plotting the extra-terrestrial irradiance values they predict.

## 2.4 $\tau_a$ intercomparison with established instrumentation

The values of $\tau_a(\lambda)$ calculated using the two spectrometers, were compared against coincidental land-based sun photometer

(CIMEL CE318, PREDE POM01-L) and marine sun photometer (Microtops II) deployments. The spectrometer hyperspectral values were integrated to give similar bandwidths (~10nm) to the sun photometers for AOD calculations. To give an accurate comparison, all the different instruments were referred to GPS time. The spectrometer datasets were filtered to select stable conditions in which $AOD_{500nm}$ varied by less than 0.05 over a 5-minute window, as measured by the spectrometer. The spectrometer filtered 1-minute readings were interpolated to the time of the comparison instrument

reading.

Spectrometer 1 (AS161) was deployed on the roof of the Plymouth Marine Laboratory (Plymouth, UK: 50° 21.95′ N, 4° 8.85′ W), in close proximity to the established ESR network (Campanelli et al., 2012) PREDE POM01-L sun photometer, between 14 July to 8 September 2014. The site is generally characterised by aerosols of a marine origin (Estelles et al., 2012). Aerosol optical properties, including $\tau_a(\lambda)$, were determined from the POM01-L measurements at 400,

500, 670, 870 and 1020nm using the inversion technique of Nakajima et al. (1996).



Spectrometer 2 (Zeiss) was deployed at the Burjassot site, which has both ESR-POM01-L and AERONET-CIMEL

CE318 sun photometers, between January and June 2016. The site is affected by many different aerosol types, including

urban, marine (Mediterranean) and Saharan dust (Estelles et al., 2007). Values of $\tau_a(\lambda)$ were determined using the CIMEL

CE318 measurements at 440, 500, 670, 870 and 1020nm, processed by AERONET algorithm version 2 (level 2 until April

2015, level 1.5 from April 2015 to June 2016). Both spectrometer systems were deployed on the Atlantic Meridional

Transect (AMT24) expedition, which sailed between the UK and the Falkland Islands on board the *RRS James Clark Ross*,

from 22 September to 4 November 2014. The transect encounters a wide variety of aerosol optical properties, from the low

$\tau_a$ background marine aerosols of the South Atlantic Ocean (Lin et al., 2016) to the higher turbidities to the west of Africa

under the influence of airborne desert dust (Caquineau et al., 2002;Baker et al., 2006). Values of $\tau_a(\lambda)$ were determined

using a manually operated handheld Microtops II instrument at 380, 440, 500, 675, 870 and 1020nm and the data processed

to level 2.0 (cloud screened, visually inspected and post-cruise calibrated) using the protocols adopted by the AERONET

Marine Aerosol Network (Smirnov et al., 2009). The estimated absolute uncertainty in individual level 2 observations does

not exceed 0.02 in any of the spectral channels.

### 3. Results

### 3.1 Improvements in measurement due to Tilt Correction

The repeated SPN1 readings give the best indication of the effectiveness of the tilt correction strategy. Detailed results are

shown for the afternoon of 30 October 2014 (Figure 5), as this was a day with relatively high pitch and roll values (peak

amplitude around 5°), and also a relatively sunny day. The time-series plot for the day shows the $I_N$ (green), $I_G$ (red) and $I_D$

(blue) values as measured directly, and the corrected $I_N$ and $I_G$ (darker colours). It is clear that the corrected values show a

large improvement for the stable clear-sky periods (e.g. 17:30 to 19:30) with the standard deviation in the readings of $I_N$

being reduced by up to a factor of four. Taking an average of the burst of SPN1 readings gives an even smoother trace, but

this option is not possible using Spectrometer 2 (Zeiss) because of the time taken to observe the entire spectrum (20s).

Figure 5 summarises this improvement by showing the standard deviation of the eight measurements within each 1

minute burst. During periods of broken cloud, variability is high. This is caused by large light level variations due to cloud

edges during the 20s burst. During clear sky periods (e.g. 17:30 to 19:30) the burst variability is reduced to 20% - 30% of the uncorrected value by implementing the correction procedure. During wholly overcast periods (e.g. 20:00 to 21:00) the variability is obviously minimised. This correction procedure is applied to all readings for Spectrometer 2 (Zeiss) during the AMT24 cruise. As a direct consequence of this, AOD values calculated from the corrected readings show less variability

during stable periods.

## 3.2 Radiometric intercomparisons

We configured the spectrometer operating software to routinely calculate four distinct datasets: (1) A daily time-series of the spectrally integrated values of global and diffuse irradiance (Figure 6). This can be presented as either an integrated $Wm^{-2}$ value across the full spectrum, or weighted by wavelength to give e.g. PAR over the range 400nm – 700nm. Other bands or

weightings can be calculated from the raw data. (2) A daily time-series of $\tau_a$ at specific wavelengths chosen to match the output of other instruments such as the Microtops II or CIMEL CE318 sun photometer. (3) Instantaneous $I_G(\lambda)$ and $I_D(\lambda)$ spectra for each measurement time (Figure **7**). (4) Instantaneous $\tau_a(\lambda)$ across the whole spectrum, outside of gaseous absorption bands, for each measurement time.

Comparisons of 1 minute spectrally integrated data from the two spectrometers with the co-located SPN1

radiometer and Kipp & Zonen PAR sensors (see Table 2 for instrument details) showed good agreement (Figure 8). PAR measurements were 4% below and 0.6% above the Kipp & Zonen PAR sensors respectively for the two spectrometers, and 26% below the SPN1 radiometer. This latter difference is largely accounted for by the different spectral ranges measured, i.e. 380nm – 1050nm for the spectrometers, 400nm – 2800nm for the SPN1. Figure 9 shows an intercomparison with the co-located Satlantic HyperSAS hyperspectral radiometer (see Table 2 for instrument details). In the range 400 – 1050 nm,

Spectrometer 1 (AS161) agrees on average within 2.3% with the HyperSAS with a maximum difference of $0.05Wm^{-2}nm^{-1}$ at 752.5nm; Spectrometer 2 (Zeiss) is within 4.7% of the HyperSAS with a maximum difference of $0.025Wm^{-2}nm^{-1}$ at 927.1nm. Spectrometers 1 and 2 are within 2.2% of each other with a maximum difference of $0.07Wm^{-2}nm^{-1}$ at 754.0nm.

## 3.3 Aerosol optical depth comparisons

*Plymouth*



Prior to the AMT24 cruise, Spectrometer 1 (AS161) was mounted on the roof at PML in Plymouth, adjacent to a PREDE POM-01 sun photometer, between 14 July – 8 September 2014. The AOD intercomparison (Figure 10) between the two instrument datasets results in a high $R^2$ (ranging between 0.768 at 870nm and 0.940 at 500nm) and an RMSE of between 0.040 (675nm) to 0.075 (400nm). This is similar to differences found between LICOR LI1800 spectrometers (Estelles et al.,

2006). The 400nm channel performance was somewhat worse than the other wavelengths using the RMSE metric (0.705). This is due largely to the diminishing sensitivity of the AS161 spectrometer at 400nm and below. There are also noticeable changes in the regression slope with wavelength in Figure 10, this varying between 0.911 (500nm) to 0.710 (870nm). The intercept value also varies between -0.012 (400nm) and 0.037 (870nm).

*Valencia*

Subsequent to the AMT24 cruise, Spectrometer 2 (Zeiss) was co-located with a CIMEL sun photometer at the Burjassot site, between January 2015 and June 2016. These land based results (Figure 11) show that there is a consistent relationship between the spectrometer and sun photometer derived AOD measurements. The regression slope varies between 0.786 at 440nm and 0.687 at 870nm (decreasing slope with increasing wavelength) with a broad decrease in the intercept from ~0.03 to 0.02 (decreasing intercept with increasing wavelength). There is also a reduction in the residuals from 0.029 at 440nm to

0.015 at 870nm. The value of $R^2$ remains largely unchanged at around 0.95. A notable feature of both Figure 10 and Figure 11 is the significant, but consistent, deviation away from the 1:1 line when comparing the different instrument retrievals of AOD. One possible source of this behaviour is thought to be the wider field-of-view (FOV) of the SPN1 optical design. This is typically between 5 - 10° whereas the POM and CIMEL instruments' FOV is ~1°. The difference between shadowband radiometer and sun photometric retrievals of AOD has previously been observed, and subsequently empirically

corrected for by di Sarra et al. (2015), and attributed to the radiant impact of aerosol forward scattering on different instrumental FOV. Here we investigate this further with a modelling study.

### 3.4 Corrections for FOV

The difference in FOV effect was investigated using the SMARTS2 (Gueymard, 2001) solar model. This has the facility for calculating the spectral $I_N$ received for different aerosol conditions and different detector FOVs. The model was run for a

range of different solar zenith angles (0 – 85 with 10° increments) and AODs (0.01 – 0.50 in 0.01 increments), and the $I_N$



calculated for a detector FOV of 7.5°, at 500nm. The AOD that would be calculated from the measured $I_N$ using the spectrometer AOD equations $1 - 8$ was compared with the AOD value input into the model (Figure 12). This shows three distinct features that are also apparent in the visual comparisons with the CIMEL (Figure 11): (1) a regression slope of approximately 0.8; (2) the generally positive Y-axis intercept and; (3) the offset is related to solar zenith angle. Further investigation also showed that introducing calibration errors to the notional 7.5° detector measurement changed the offsets due to solar zenith angle, and spreads the lines of different solar zenith angle further apart.

Using these insights from modelling, we are able to give a much closer correspondence to the Valencia CIMEL CE318 by: (1) using the calibration transferred from the CIMEL CE318 for all values, rather than the original (Mt. Teide) Langley calibration. The calibration adjustment for wavelength values between the CIMEL CE318 channel values is done using a linear interpolation; (2) applying a correction function for each CIMEL wavelength consisting of an offset related to solar zenith angle (air mass), then a further linear transformation in AOD to give a true estimate of AOD as measured by the CIMEL CE318. The calculated correction factors (Table 3) are selected to give the best fit to the CIMEL CE318 AOD values and applied using an equation of the form:

$$AOD(\lambda)_{corr} = (AOD(\lambda)_{meas} - OffsetA(airmass) - OffsetW(\lambda)) \times SlopeW(\lambda) \quad (15)$$

These corrections show an RMSE of 0.02 to 0.03 when compared with the CIMEL CE318 (Figure 13). While not perfect, this is approaching the uncertainty of AERONET field deployed CIMEL instruments (0.01-0.02) and the level of agreement between different sun photometers when they are compared together in the field (0.01-0.02) using different AOD methodologies (Estelles et al., 2006). LICOR 1800 spectroradiometers calibrated by lamps also have a nominal AOD uncertainty of about $0.02 - 0.05$ (Estelles et al., 2006). These corrections were then applied to the Spectrometer 2 results from the AMT cruise.

**3.5 AMT Cruise**

During the AMT cruise, Microtops readings were taken when the sky was deemed sufficiently clear (clear view of the solar disc unobscured by clouds), and research schedules permitted time. Figure 14 shows these results plotted against latitude for the entire cruise for both Spectrometer 2 (Zeiss) and the Microtops. The spectrometer results have been corrected for using the values determined using the 18 month intercomparison at the Burjassot site (Figure 11 and Figure 13 and Table 3).



Background marine aerosol ($AOD_{500nm}$ < 0.05) values are apparent in the region around 40°N and between 20°S and 40°S. Elevated values of AOD are clearly visible in locations associated with the Saharan dust plume (20°N: $AOD_{500nm}$ ~0.5) and European anthropogenic pollution emitted by a combination of industrial and urban sources (50°N: $AOD_{500nm}$ ~0.4). Comparisons between the Spectrometer 2 (Zeiss) and Microtops at four different wavelengths (440, 500, 675 and 870nm –

Figure 15) results in a RMSE between 0.04 and 0.05, which is poorer than the results presented against the CIMEL (RMSE ~ 0.03: Figure 11). However, the coefficient of determination ($R^2$) remains high at around 0.95 for all wavelengths. Previous (unpublished) comparisons between Microtops and CIMEL CE318 resulted in an RMSE of between 0.01 – 0.02; an agreement to within 10% between Microtops, CIMEL and POM instruments has been reported in Poland under a variety of conditions (Evgenieva et al., 2008).

**4. Discussion**

Overall the hyperspectral radiometers that we have developed gave excellent and robust performance in the field (terrestrial and marine) over protracted periods of deployment, with little or no operator intervention. The marine deployment in particular highlighted previously unforeseen practical issues. These were to do with shading and soiling of the detector dome. While the instrument platform on the *RRS James Clark Ross* gave a reasonably good exposure to the sky, there was

some shading possible, in particular by two higher masts just forward of the spectrometers, containing the HyperSAS hyperspectral radiometer, and an ultrasonic anemometer. The meteorological instrument solar radiation screens and ship's main mast on the bridge could also obstruct the sun when close to the horizon. It was possible to identify and filter out many of these obstruction periods by comparing the outputs of adjacent sensors. In principle, it should also be possible to predict these occasions using a combination of the solar geometry, position and height of the masts relative to the instrument, and

the ships attitude. However, this has not been done in this paper. There is always intense competition for the 'top spot' on any ship, so some form of shading at times is always likely to be a problem.

Access to the instrument platform was restricted during the AMT24 cruise, so it was not possible to inspect or clean any of the instruments. The position of the mast towards the bow of the ship also brought it closer to birds slip-streaming the forward air-pressure wave as well as providing a good position for perching. The instrument platform itself showed





evidence of many direct hits from bird droppings and there was white residue from fouling on the dome of Spectrometer 1 discovered upon instrument retrieval at the end of the cruise. This will have obviously caused degradation in the signal intensity. Therefore finding a position on the ship superstructure enabling a complete and unobstructed view of the sky as well as allowing access for periodic cleaning would improve data quality. Multiple, season long deployments (6 – 12

months) of the SPN1 on the Western Channel Observatory buoy at station L4 (Smyth et al., 2010), have shown the instrument remarkably resilient to such problems though, as it is always retrieved in a pristine condition. It is likely here that regular washing by rainwater keeps the dome free from fouling.

        The storing and processing of the quantity of data produced by each spectrometer (100Mb/day Spectrometer 1; 30Mb/day Spectrometer 2 for one minute readings) is a significant task. In order to report readings back via satellite Iridium

communications, enabling full autonomy on ships of opportunity, will either require a significant amount of data compression, or a limited subset of measurements to be reported back. Full datasets, allowing in-depth analysis and quality control, will only be retrievable upon the completion of individual deployments. Therefore, further development is required to provide a balance between reporting derived quantities such as AOD, and retaining the raw measurements to allow for further corrections or new products later.

The field intercomparisons of AOD carried out in this paper with existing multi-spectral instrumentation, have necessarily been restricted to wavelengths at 400, 440, 500, 670 and 870 nm. However, as both Spectrometer 1 and 2 are hyperspectral instruments, retrieval of hyperspectral AOD observations are theoretically possible. To fully enable this more work is required on the calibration of the instrument (where direct transferability between standard instruments is no longer possible) and correction for gaseous absorption (e.g. $NO_x$, $O_2$ and $H_2O$ absorption bands).

The other limiting factor in this paper has been in the time-dimension. Handheld Microtops measurements are generally taken on an opportunistic footing, when a dedicated operator is not available; CIMEL and POM measurements are generally taken on a 10 – 15 minute time interval. As observed by di Sarra et al. (2015), the shadowband type technology can take readings on a sub-minute timescale, which allows almost continuous observations of AOD and the resolution of short length and time-scale atmospheric aerosol features and variability. Although placing Spectrometer 1 and 2 on a ship,





with many other sources of error such as motion and variable ship shading, may preclude accurate observation of such features, a land-based deployment should allow this opportunity.

## 5. Conclusions

The hyperspectral radiometer that we have developed and described in this paper has many advantages over the current generation of sun / sky radiometers. The system has the potential for operating remotely and autonomously for long-periods of time on ships of opportunity. As it has no moving parts, shade rings or motorised tracking it lessens the number of points of failure which are particularly vulnerable in the marine environment (salt corrosion, freezing temperatures).

The fieldwork components of this study highlighted many issues which needed resolving. Some of these have been resolved such as correcting for the motion of the ship; other issues such as characterisation and calibration have been partially resolved. The calibration issue is crucial and the use of a Langley method as well as suitable periods of time using co-located instrumentation which are traceable to standards is required. This is standard within the existing networks such as AERONET (Holben et al., 1998). The development of a fully robust calibration protocol for the complete spectral range still requires development, together with a test of the correction (FOV and solar zenith angle) algorithms under a wider range of conditions than has been possible in this paper. The aerosol forward scatter / FOV difference issue has been partially resolved using both theory and field measurements. However, the correction coefficients are likely to be specific to individual instruments. Overall, this paper has shown the technology that we have developed, together with its associated algorithms, to be a viable option when considering instrumentation for deployment on ships of opportunity in supporting and widening the global AERONET, SKYNET and ESR networks in the data sparse expanses of the ocean. The technology should also be transferrable to satellite calibration and validation studies, enabling the development of moveable fiducial points if deployed on e.g. an autonomous platform such as a waveglider.

## Data availability

All data used in the generation of this manuscript is available on request from TS.



**Team list**

John Wood (JW), Tim Smyth (TS) and Víctor Estellés (VE).

**Author contribution**

JW built the two spectrometer systems described within this manuscript, carried out the calibrations and data analysis, and contributed to the writing of this manuscript. TS took the two spectrometer systems on the AMT24 cruise and made all the ancillary measurements using the Microtops sun photometer, oversaw the Plymouth intercomparison fieldwork, undertook the modelling work and compiled the writing of this manuscript. VE oversaw the Valencia intercomparison fieldwork and contributed to the writing of this manuscript.

**Competing interests**

The authors declare that they have no conflict of interest.

**Disclaimer**

**Acknowledgements**

The authors would like to thank the crew of the *RRS James Clark Ross* and staff of the British Antarctic Survey for the considerable help they gave in deploying the spectrometers during AMT24. JW would like to thank Delta-T Devices Ltd for their encouragement, and the East Midlands Development Agency for funding assistance during the early stage of prototype design. VE was funded by the European Regional Development Fund, the Spanish Ministry of Economy and Competitiveness (CGL2015-64785-R, CGL2015-70432-R), and the Valencia Autonomous Government (PROMETEUII/2014/058). The Microtops data collected on AMT24 were processed by the NASA AERONET Maritime Aerosol Network program. This study is a contribution to the international IMBER project and was supported by the UK Natural Environment Research Council National Capability funding to Plymouth Marine Laboratory (TJS) and the National Oceanography Centre, Southampton. This is contribution number 307 of the AMT programme.

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





## Figures

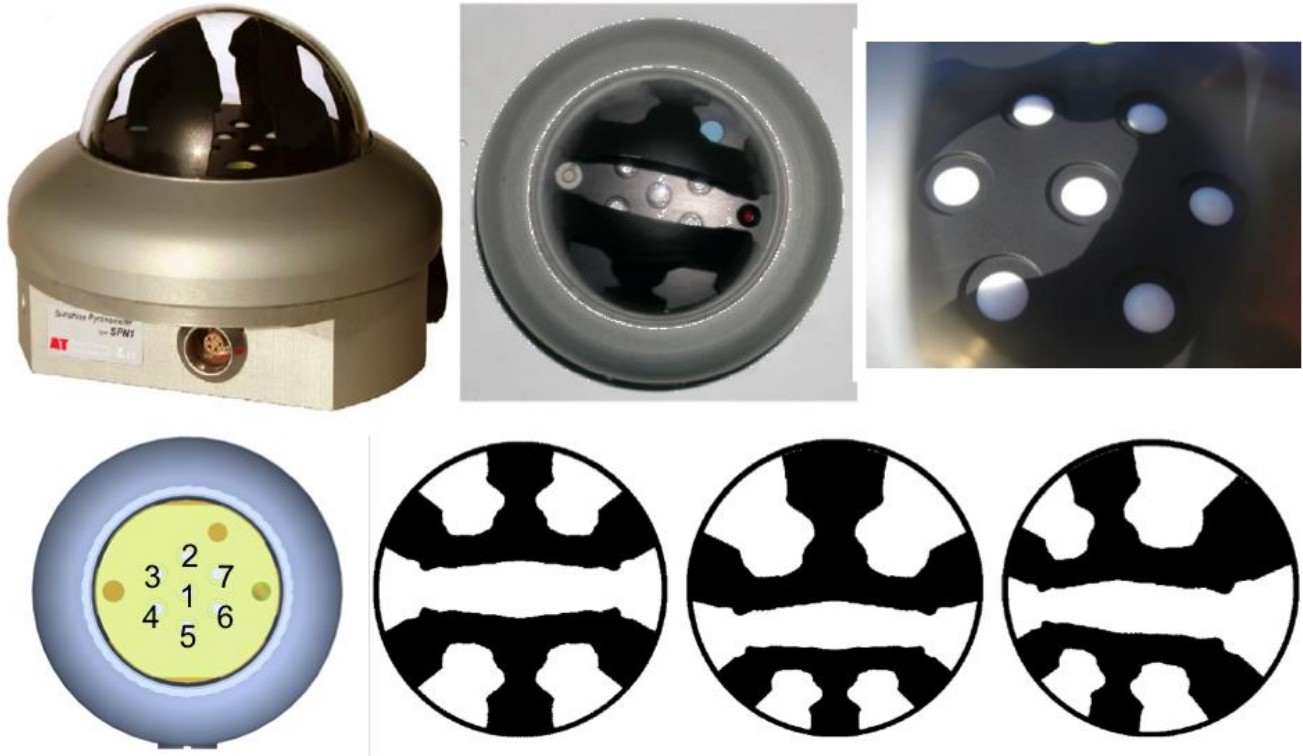

**Figure 1: Taken from Badosa et al. (2014). Top row: Left photo shows the side view of the SPN1 and the middle is a photograph from directly above the unit. Photo on the right demonstrates the shadow pattern on the seven sensors under direct sunshine conditions. Bottom row: Left gives SPN1 detector numbering; sky seen under shade patterns as seen for sensor 1 (left), sensors 2 and 5 (middle) and sensors 3, 4, 6 and 7 (right).**





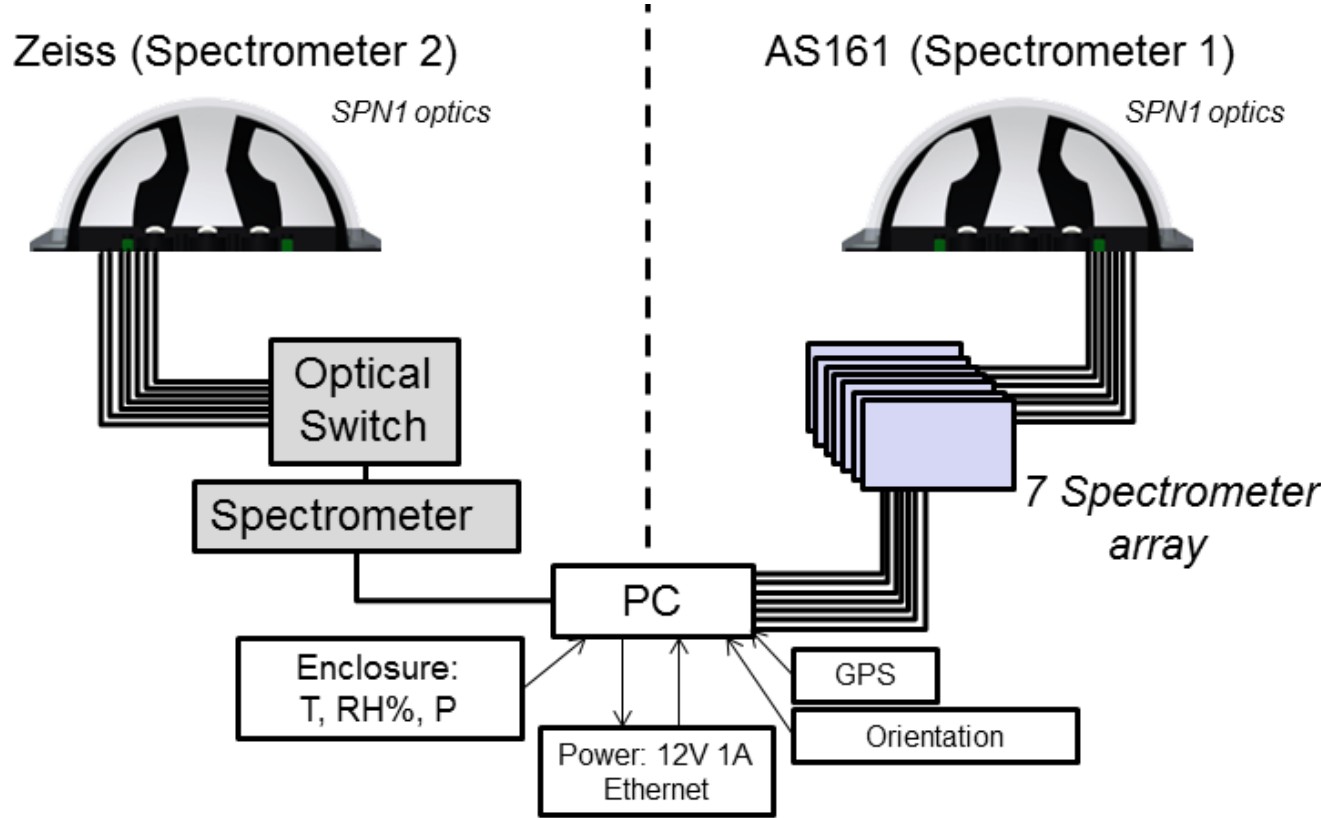

**Figure 2: System diagram for the two spectrometer configurations. Elements in white are common to both configurations, although each have their own separate PC, GPS etc. Main configurational difference is that the AS161 (Spectrometer 1) contains seven spectrometers, whereas the Zeiss (Spectrometer 2) contains only one which is connected to the seven optical channels via an**
5  **optical switch. The PC enclosure temperature (T), relative humidity (RH) as well as atmospheric pressure (P) is monitored.**



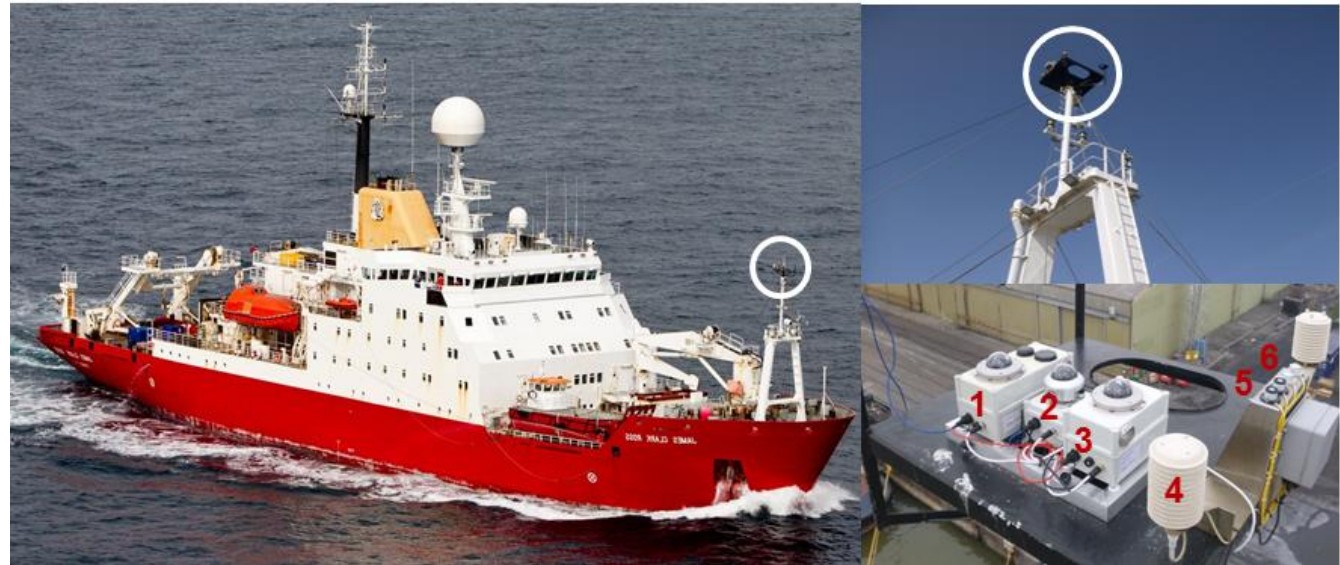

**Figure 3: Main image - RRS James Clark Ross showing the position of the foremast and instrument platform (circled). Top right: The instrument platform (circled) viewed from below on the main-deck. Bottom right: instruments in situ on the platform. (1) Spectrometer 1 – AS161; (2) SPN1; (3) Spectrometer 2 - Zeiss; (4) meteorological instrument solar radiation shield; (5) Kipp & Zonen PAR sensors (x2); (6) Kipp & Zonen pyranometers (x2).**




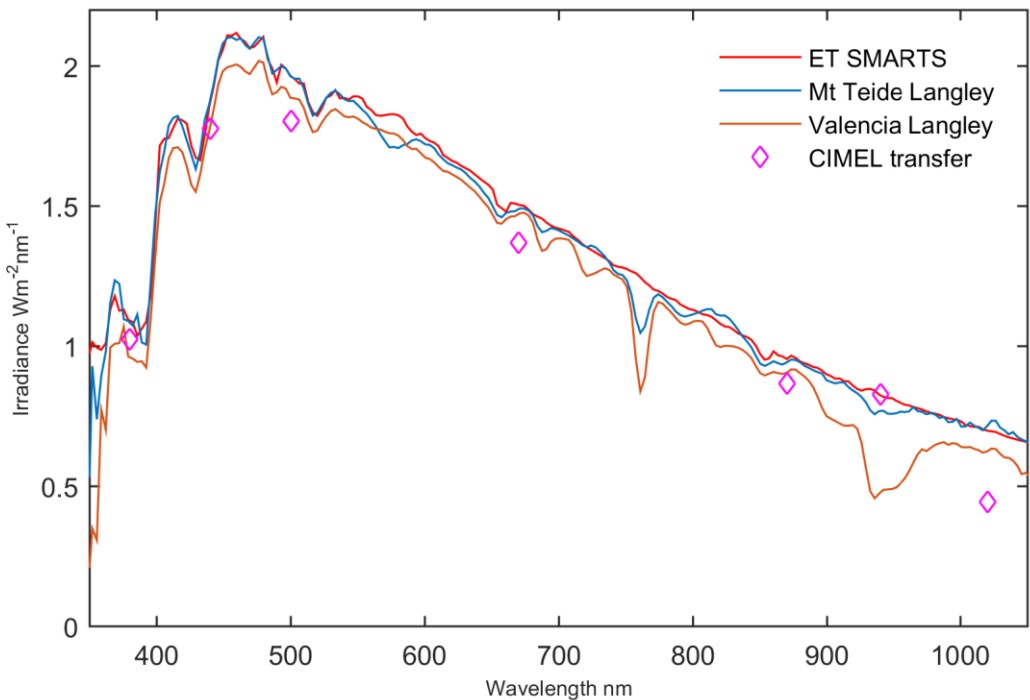

**Figure 4: Calibration curve for Spectrometer 2 (Zeiss). Extra-terrestrial spectrum as predicted from the SMARTS2 model, Langley calibrations on Mt Teide (Tenerife), Valencia (Spain), and from calibration transfer from the CIMEL sun photometer at Valencia.**





**Figure 5: SPN1 tilt correction illustration data for 30 October 2014. a) Uncorrected $I_N$ (light green) and corrected $I_N$ (dark green), Uncorrected and corrected $I_G$ (light and dark red) together with $I_D$ (blue). b) Standard deviation for uncorrected $I_N$ (red) and corrected $I_N$ (blue).**







**Figure 6: Time-series plots of integrated PAR values of $I_G$ and $I_D$ (upper plot) and AOD$_{500nm}$ (lower plot) for 4 October 2014. The Microtops 500nm values are superimposed on the AOD plot. Cloud affected AOD have not been removed from the spectrometer database.**





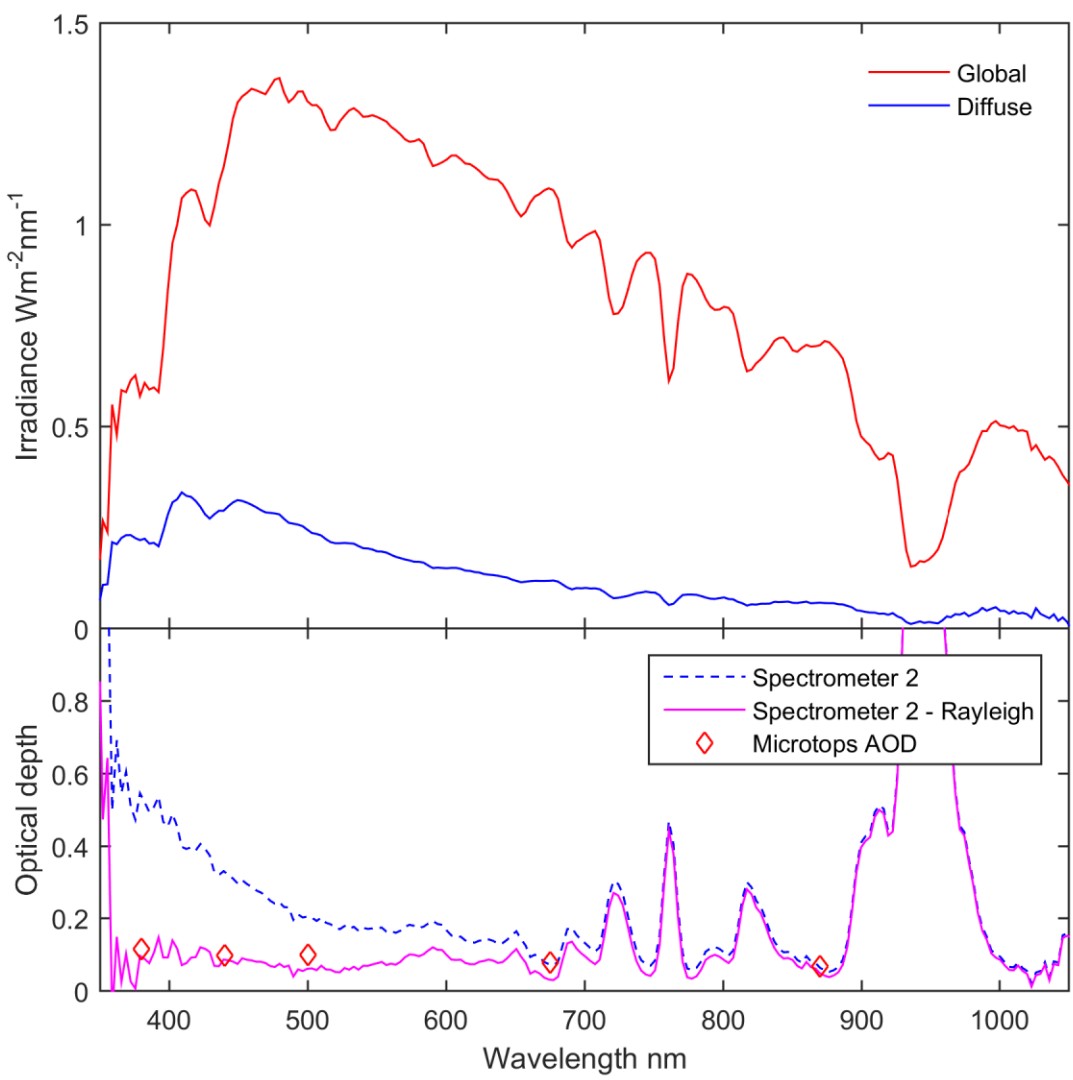

**Figure 7: Spectrally resolved outputs for a single reading (12:00 on 4 October 2014). $I_G$ and $I_D$ spectra (upper plot), and Optical Depth (lower plot). Total Optical Depth (OD) and OD with the Rayleigh component removed are shown, with Microtops Aerosol Optical Depth (AOD) values superimposed. Gaseous absorption features at certain windows have not been removed.**







**Figure 8: Comparisons of integrated PAR (400nm – 700nm, quantum weighting) and Energy (integrated over 380nm – 1050nm) with the adjacent Kipp & Zonen PQS-1 PAR sensor, and SPN1.**




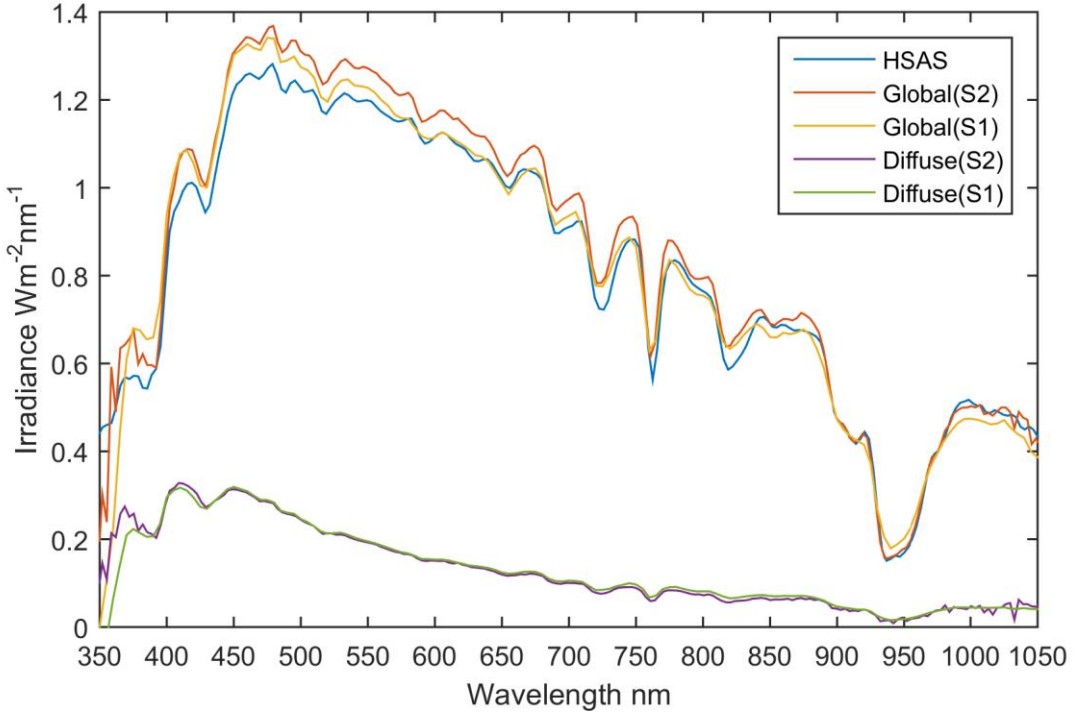

**Figure 9: Spectral outputs ($I_G$ and $I_D$) from the two spectrometers compared with the HyperSAS $I_G$ at 12:00 on 4 October 2014.**







**Figure 10: Spectrometer 1 (AS161) AOD results compared with PREDE POM-01 on the roof of Plymouth Marine Laboratory, 14 July – 8 September 2014. Spectrometer readings restricted to clear stable periods. These are log density plots: red points represent around 100 data points, whereas the blue points only a single data point. No further corrections applied.**



**Figure 11: Spectrometer 2 (Zeiss) AOD results compared with CIMEL sun photometer at Burjassot, January 2015 – July 2016. These are log density plots: red points represent around 100 data points, whereas the blue points only a single data point. No further corrections applied.**



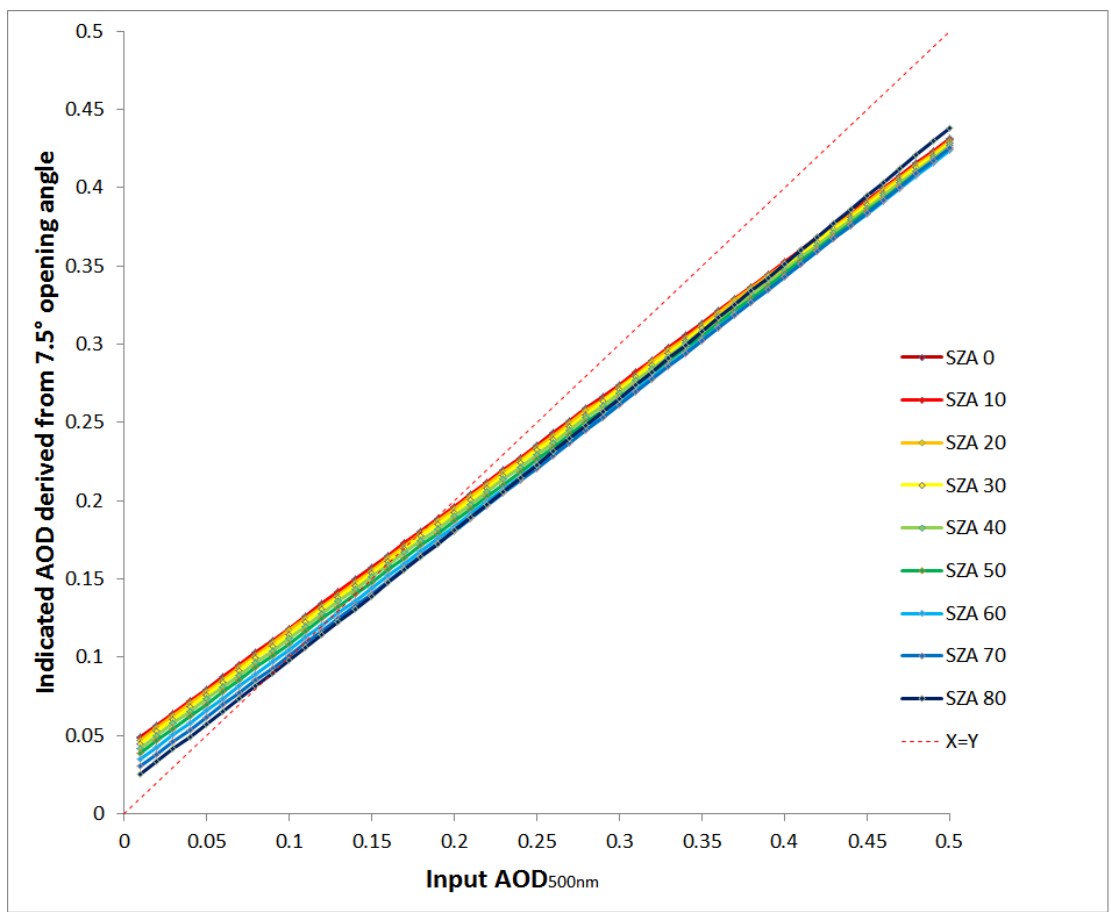

**Figure 12: Theoretical AOD computed from measured $I_N$ according to the SMARTS2 (Gueymard, 2001) model, when using a 7.5°
FOV detector. The different coloured lines represent different solar zenith angles. Indicated AOD instead of AOD to be consistent
with the text**




**Figure 13: Corrected Zeiss AOD values compared with CIMEL sun photometer at Burjassot.**





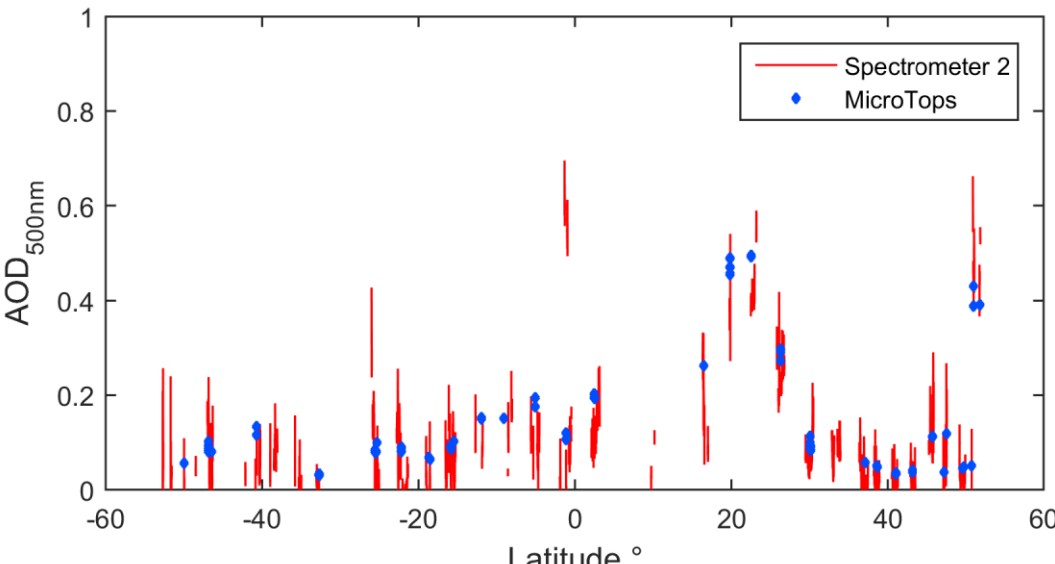

**Figure 14: General evolution of AOD$_{500nm}$ as measured by Spectrometer 2 (Zeiss, red line) and Microtops (blue diamonds) over the duration of the AMT24 cruise.**





**Figure 15: Comparison of Spectrometer 2 (Zeiss) and Microtops AOD measurements at four wavelengths over the AMT cruise.**



| Symbol | Description | SI Units |
|---|---|---|
| $I_G(\lambda)$ | (Spectral) Global Horizontal Irradiance (GHI) | $Wm^{-2}(nm^{-1})$ |
| $V_G(\lambda)$ | (Spectral) GHI measured as a voltage | V |
| $I_N(\lambda)$ | (Spectral) Direct Normal Irradiance (DNI) | $Wm^{-2}(nm^{-1})$ |
| $V_N(\lambda)$ | (Spectral) DNI measured as a voltage | V |
| $I_D(\lambda)$ | (Spectral) Diffuse Horizontal Irradiance (DHI) | $Wm^{-2}(nm^{-1})$ |
| $V_D(\lambda)$ | (Spectral) DHI measured as a voltage | V |
| $I_H(\lambda)$ | (Spectral) Direct Beam Horizontal Irradiance (BHI) | $Wm^{-2}(nm^{-1})$ |
| $V_H(\lambda)$ | (Spectral) BHI measured as a voltage | V |
| $V_T(\lambda)$ | (Spectral) Top of Atmosphere (TOA) voltage | V |
| $V_{0T}(\lambda)$ | (Spectral) TOA voltage corrected for elliptical Earth orbit | V |
| $I_{max}(\lambda)$ | (Spectral) Maximum irradiance | $Wm^{-2}(nm^{-1})$ |
| $I_{min}(\lambda)$ | (Spectral) Minimum irradiance | $Wm^{-2}(nm^{-1})$ |
| $\theta_{rs}$ | Relative solar angle (angle of incidence to plane of detector) | Radians |
| $\theta_s$ | Solar zenith angle | Radians |
| $\alpha_{sf}$ | Surface zenith angle | Radians |
| $\varphi_s$ | Solar azimuth angle | Radians |
| $\beta_{sf}$ | Surface azimuth angle | Radians |
| $\tau_a(\lambda)$ | Aerosol Optical Depth | Unitless |
| $\tau_R(\lambda)$ | Rayleigh Optical Depth | Unitless |
| $\tau_O(\lambda)$ | Ozone Optical Depth | Unitless |

**Table 1: Glossary of terms and symbols**





| Instrument | Serial # | Description | Calibration dates and details | Intercomparison dates and details |
|---|---|---|---|---|
| Kipp & Zonen PQS 1 | 110126 and 110127 | Kipp & Zonen PAR sensors for 400 – 700 ± 4 nm range situated on RRS James Clark Ross instrument platform. | Kipp & Zonen factory calibration against known standards 05/01/2011 | 22/09/2014 – 01/11/2014 AMT24 against calculated integrated PAR from Spectrometer 1 and 2. |
| Kipp & Zonen SP-Lite | 112992 & 112993 | Kipp & Zonen Energy sensors for 400 – 1100 nm range situated on RRS James Clark Ross instrument platform. | Kipp & Zonen factory calibration 26/01/2011 | 22/09/2014 – 01/11/2014 AMT24 against calculated integrated Energy from Spectrometer 1 and 2. |
| Satlantic HyperSAS hyperspectral radiometer | SATHSE0258 | Satlantic hyperspectral irradiance sensor for 305 – 1142 nm range at 3 nm resolution. Situated on RRS James Clark Ross instrument platform. | Satlantic factory calibration against known standards 06/01/2014 | 22/09/2014 – 01/11/2014 AMT24 against hyperspectral data from Spectrometer 1 and 2. |
| Spectrometer 1 | AS161 | See text for details. Situated on RRS James Clark Ross instrument platform. | Laboratory calibration (02/10/2012) at Winster; Field calibration (Langley) (25/06/2014) at Mt Teide; | 14/07/2014 – 08/09/2014 against POM-01 at Plymouth PML 22/9/14 – 01/11/14 AMT24 cruise |
| Spectrometer 2 | Zeiss | See text for details. Situated on RRS James Clark Ross instrument platform. | Laboratory calibration (11/03/2014) at Winster; Field calibration (Langley) (25/06/2014) at Mt Teide; Field calibration adjustment at Burjassot against CIMEL #953 using 11 clear-sky days 17/05/2015 to 01/06/2016 | 22/9/14 – 01/11/14 AMT24 cruise 29/01/2015 – 09/06/2016 against CIMEL 318 at Burjassot. |
| SPN1 | A749 | Delta-T broadband Global & Diffuse energy sensor. Situated on James Clark Ross instrument platform. | Field calibration at MeteoSwiss Payerne solar measurement station Jun 2012 – Sept 2013 | 22/9/14 – 01/11/14 AMT24 cruise |

**Table 2: Description and calibration details of instruments used in this manuscript**





| Airmass | 1 | 2 | 3 | 6 | 10 |
|---|---|---|---|---|---|
| **OffsetA** | 0.0097 | 0.0177 | -0.0033 | -0.0067 | -0.0117 |

| Wavelength (nm) | 440 | 500 | 675 | 870 | 1020 |
|---|---|---|---|---|---|
| **OffsetW** | 0.0244 | 0.026 | 0.0182 | 0.0124 | 0.0457 |
| **SlopeW** | 1.2701 | 1.2893 | 1.3549 | 1.4522 | 1.5237 |

Table 3: Correction values applied to AOD measured using Spectrometer 2. The correction factor is applied using equation 15.