# Peer review of "Autonomous marine hyperspectral radiometers for determining solar irradiances and aerosol optical properties"

_Atmospheric Measurement Techniques, 2016_

## Referee Comment (RC1) · Anonymous Referee #1 · 23 Feb 2017

The manuscript deals with the development of two hyperspectral radiometers specially suited to be deployed in in harsh and remote environments. Such instruments would make the determination of the aerosol optical depth easier over the oceans, where the coverage provided by the existing network is quite scarce.

The overall scientific quality is very good, including the comparison with other existing instruments, and the issues which still need resolving are adequately addressed.

To my opinion, the manuscript should be accepted as is.

---

## Referee Comment (RC2) · Anonymous Referee #3 · 28 Feb 2017

The paper describes an interesting and useful development of a commercial instrument for simultaneous monitoring of global and diffuse spectral irradiances, allowing for the determination of the aerosol optical depth. The paper provides a description of the device, with two different setups, and a first evaluation of the collected data in an oceanic cruise and during groud-based measurements.

The instruments appear to perform well under different conditions, and the proposed solution is particulary useful since it permits the automated acquisition of spectral global and diffuse irradiances, and to retrieve spectral aerosol optical depths. The same device, in correspondance with absorption bands of atmospheric gases, might also be used to determine column amounts of different atmospheric species.

[Figure]

The paper deserves publication; however, the following aspects should be improved:

1. the overall organization of the paper, which seems to somewhat mix up different things

2. the description and treatment of the measurement uncertainties.

3. the description of the instrumental characteristics and calibration procedures.

Regarding the first point, I would suggest discussing in separate chapters: i. the technological implementation (sections 2.2.1-2.2.5, 2.2.7-2.2.8); ii. the description of the measurement sites and setup; iii. corrections for ship motion and FOV to AOD measurements; iv. intercomparison of corrected AODs and irradiances.

Secondly, the discussion of the uncertainties may be improved (see, e.g., Miller et al., 2004); for instance, a discussion of the role of the instrument temperature dependence, cosine response, response time, as well as the effect of uncertainties on the instrument attitude (influence of angle uncertainty on the tilt angle correction) should be added. In the same context, the authors are using calibrations, or measurements with different instruments referring to calibrations, performed over a wide temporal interval (2011-2016). Possible effects due to instrumental long-term drifts should be discussed.

As a third point, the authors should provide additional information on the spectral resolution, in addition to the pixel resolution, spectral stability, and temperature dependence of the two spectrometers. Similarly, the main characteristics of the HyperSAS instrument should be included. Measurements uncertainties on PAR and global shortwave irradiances should also be reported.

Minor comments follow.

page 4, line 9: to my knowledge, the first application of the rotating shadowband technique to AOD measurements on ships is by Guzzi et al. (1985).

p.4, l. 10: the discussion is valid only for instruments with an ideal cosine responsivity

of the input optics

p.5, l. 16: this is one of different possible relationships for the airmass

p.7, l. 18: what is the spectral resolution of the spectrometer? Is there any information on long-term stability, temperature dependence, and stray light?

p.8, l. 3: same as above, for the Zeiss spectrometer.

p.11, l. 5-6: how is the lamp calibration with the integrating sphere made? What is intended for "approximately correct overall calibration"? Which lamp types are used? Please, add details.

p. 11, l. 9-11: figure 4 shows extraterrestrial irradiances derived with the Langley plot method throughout the spectrum; the method is not directly applicable in correspondence with absorption lines/bands. For instance, the value retrieved from the Langley plot method is not expected to correspond with the extraterrestrial irradiance in particular in the 940 nm water vapour band.

p. 11, l. 11-12: differences in figure 4 appear to be between few and about 20%, with large differences mainly at 1020 nm. What is the estimated uncertainty on the different determinations? May those be added to the graph? May the time different between calibrations have a role? These differences in the extraterrestrial values are expected to produce a significant impact on the retrieved AOD values. Which extraterrestrial values have been used in the analysis?

p.12, l. 5: "Both spectrometer systems...": please, start a new paragraph

p. 13, l. 4: please, specify where these data have been acquired.

p.13, l. 18-22: please, provide information on the HyperSAS spectral resolution. What are the estimated uncertainties on the measured irradiances? Please, note that largest diferences occur within absorption bands; different spectral resolutions may play a role here.

p.14, l. 2-4, and figure 10: the linear regression seems to be strongly influenced by few data points with large POM AOD and small Spectrometer AOD, especially at 675 and 870 nm; did the authors try to identify and understand why there are large differences between POM and the spectrometer for these points? Is there a reason for the exclusion of data at 1020 nm?

p.14, l. 11-14: the relationship seems to be non-linear (figure 11).

p.15, l. 4-6: "... introducing calibration errors to the notional 7.5° detector measurement...": the sentence is not clear. What is author's best estimate of the dependency on solar zenith angle? Is it negligible? If it is not, the correction scheme should take into account the solar zenith angle. Maybe I miss something, but it is not clear to me why the simulations produce a positive Y-axis intercept, since a larger FOV always implies an overestimate of the direct component. Do the authors have an explanation for this?

p.15, l. 24, and figure 14: it may be helpful to add the corresponding longitude on the upper X axis, or a map of the ship track. Which is the frequency of Microtops measurements? Are the data single measurements, daily/latitude averages? Please, specify.

p.16, l. 4-6: is not this difference in the RMSE expected? Data in figure 11 are on the ground, and no uncertainties due to the platform motion are present. Moreover, the Microtops AOD has a somewhat larger uncertainty than Cimel.

p.16, l. 13: shading, cleaning, and soiling effects were not discussed previously in the text; how and how many many data have been discarded? Can some of the data affected by these effects be identified in the scatterplots?

References

Guzzi, R., G.C. Maracci, R. Rizzi, and A. Siccardi, Spectroradiometer for ground-based atmospheric measurements related to remote sensing in the visible from a satellite,

Appl. Opt., 24, 2859-2864, 1985.

Miller, M.A., M. J. Bartholomew, and R. M. Reynolds, The accuracy of marine shadow-band Sun photometer measurements of aerosol optical thickness and Angstrom exponent, J. Atmos. Oceanic Tech., 21, 397-410, 2004.

---

## Author Comment (AC1) · 4 Apr 2017

We thank the anonymous reviewer for their support for our paper.

---

## Author Comment (AC2) · 4 Apr 2017

We thank reviewer #3 for their thorough examination of our paper. We have hopefully addressed the points that were raised. We particularly thank the reviewer for highlighting the discrepancy between the field-of-view (FOV) modelling calculations and what should be expected in reality. As a result of this we re-ran the calculations and found residual issues with calibration can cause the offsets observed, rather than anything to do with viewing geometry.

Please find attached a zip archive containing the following pdf files: (1) The points raised by the reviewer (Q1 - 20) followed by our response (R1 - 20); (2) A track changes version of the paper.

[Figure]

Please also note the supplement to this comment:
http://www.atmos-meas-tech-discuss.net/amt-2016-373/amt-2016-373-AC2-
supplement.zip

———————————————————

---

## Author Response (AR1)

Points raised by Anonymous Referee #3

*Author: The reviewers questions have been numbered Q3.1 – Q3.20. The corresponding responses to these questions are logically numbered R3.1 – R3.20.*

The paper describes an interesting and useful development of a commercial instrument for simultaneous monitoring of global and diffuse spectral irradiances, allowing for the determination of the aerosol optical depth. The paper provides a description of the device, with two different setups, and a first evaluation of the collected data in an oceanic cruise and during ground-based measurements.

10  The instruments appear to perform well under different conditions, and the proposed solution is particularly useful since it permits the automated acquisition of spectral global and diffuse irradiances, and to retrieve spectral aerosol optical depths. The same device, in correspondence with absorption bands of atmospheric gases, might also be used to determine column amounts of different atmospheric species.

The paper deserves publication; however, the following aspects should be improved:

15  1. the overall organization of the paper, which seems to somewhat mix up different things

2. the description and treatment of the measurement uncertainties.

3. the description of the instrumental characteristics and calibration procedures.

**(Q3.1)** Regarding the first point, I would suggest discussing in separate chapters: i. the technological implementation (sections 2.2.1-2.2.5, 2.2.7-2.2.8); ii. the description of the measurement sites and setup; iii. corrections for ship motion and

20  FOV to AOD measurements; iv. intercomparison of corrected AODs and irradiances.

**(Q3.2)** Secondly, the discussion of the uncertainties may be improved (see, e.g., Miller et al., 2004); for instance, a discussion of the role of the instrument temperature dependence, cosine response, response time, as well as the effect of uncertainties on the instrument attitude (influence of angle uncertainty on the tilt angle correction) should be added. In the same context, the authors are using calibrations, or measurements with different instruments referring to calibrations,

25  performed over a wide temporal interval (2011-2016). Possible effects due to instrumental long-term drifts should be discussed.

**(Q3.3)** As a third point, the authors should provide additional information on the spectral resolution, in addition to the pixel resolution, spectral stability, and temperature dependence of the two spectrometers. Similarly, the main characteristics of the HyperSAS instrument should be included. Measurements uncertainties on PAR and global shortwave irradiances should also

30  be reported.

Minor comments follow.

**(Q3.4)** page 4, line 9: to my knowledge, the first application of the rotating shadowband technique to AOD measurements on ships is by Guzzi et al. (1985).

**(Q3.5)** p.4, l. 10: the discussion is valid only for instruments with an ideal cosine responsivity of the input optics

**(Q3.6)** p.5, l. 16: this is one of different possible relationships for the Airmass

**(Q3.7)** p.7, l. 18: what is the spectral resolution of the spectrometer? Is there any information on long-term stability, temperature dependence, and stray light?

**(Q3.8)** p.8, l. 3: same as above, for the Zeiss spectrometer.

**(Q3.9)** p.11, l. 5-6: how is the lamp calibration with the integrating sphere made? What is intended for "approximately correct overall calibration"? Which lamp types are used? Please, add details.

**(Q3.10)** p. 11, l. 9-11: figure 4 shows extraterrestrial irradiances derived with the Langley plot method throughout the spectrum; the method is not directly applicable in correspondence with absorption lines/bands. For instance, the value retrieved from the Langley plot method is not expected to correspond with the extraterrestrial irradiance in particular in the 940 nm water vapour band.

**(Q3.11)** p. 11, l. 11-12: differences in figure 4 appear to be between few and about 20%, with large differences mainly at 1020 nm. What is the estimated uncertainty on the different determinations? May those be added to the graph? May the time different between calibrations have a role? These differences in the extraterrestrial values are expected to produce a significant impact on the retrieved AOD values. Which extraterrestrial values have been used in the analysis?

**(Q3.12)** p.12, l. 5: "Both spectrometer systems...": please, start a new paragraph

**(Q3.13)** p. 13, l. 4: please, specify where these data have been acquired.

**(Q3.14)** p.13, l. 18-22: please, provide information on the HyperSAS spectral resolution. What are the estimated uncertainties on the measured irradiances? Please, note that largest differences occur within absorption bands; different spectral resolutions may play a role here.

**(Q3.15)** p.14, l. 2-4, and figure 10: the linear regression seems to be strongly influenced by few data points with large POM AOD and small Spectrometer AOD, especially at 675 and 870 nm; did the authors try to identify and understand why there are large differences between POM and the spectrometer for these points? Is there a reason for the exclusion of data at 1020 nm?

**(Q3.16)** p.14, l. 11-14: the relationship seems to be non-linear (figure 11).

**(Q3.17)** p.15, l. 4-6: "... introducing calibration errors to the notional 7.5 detector measurement...": the sentence is not clear. What is author's best estimate of the dependency on solar zenith angle? Is it negligible? If it is not, the correction scheme should take into account the solar zenith angle. Maybe I miss something, but it is not clear to me why the simulations produce a positive Y-axis intercept, since a larger FOV always implies an overestimate of the direct component. Do the authors have an explanation for this?

**(Q3.18)** p.15, l. 24, and figure 14: it may be helpful to add the corresponding longitude on the upper X axis, or a map of the ship track. Which is the frequency of Microtops measurements? Are the data single measurements, daily/latitude averages? Please, specify.

**(Q3.19)** p.16, l. 4-6: is not this difference in the RMSE expected? Data in figure 11 are on the ground, and no uncertainties due to the platform motion are present. Moreover, the Microtops AOD has a somewhat larger uncertainty than Cimel.

**(Q3.20)** p.16, l. 13: shading, cleaning, and soiling effects were not discussed previously in the text; how and how many data have been discarded? Can some of the data affected by these effects be identified in the scatterplots?

References

Guzzi, R., G.C. Maracci, R. Rizzi, and A. Siccardi, Spectroradiometer for ground-based atmospheric measurements related to remote sensing in the visible from a satellite, Appl. Opt., 24, 2859-2864, 1985.

Miller, M.A., M. J. Bartholomew, and R. M. Reynolds, The accuracy of marine shadowband Sun photometer measurements of aerosol optical thickness and Angstrom exponent, J. Atmos. Oceanic Tech., 21, 397-410, 2004.

**Response to Reviewer #3**

**(R3.1)** We have extensively reordered the manuscript to reflect the reviewers concerns. It is now broadly grouped around the following (taken verbatim from the end of the Introduction):

"A methods section (Section 2) describing the theoretical basis (Section 2.1) and technological implementation (Section 2.2) of our approach together with the field-site setup (Section 2.3) and instrument calibration (Section 2.4). The results section (Section 3) focusses on correcting the measurements for orientation (Section 3.1) and field-of-view differences (Section 3.2) tackled using theoretical and land-based intercomparison campaigns; an intercomparison with co-located established marine radiometric instrumentation (Section 3.3) and finally an intercomparison with marine field measurements of aerosol optical depth corrected for orientation and field-of-view (Section 3.4)."

**(R3.2)** The discussion section has been updated. We thank the reviewer for bringing the Miller et al. (2004) paper to our attention as it provides a useful framework for error / uncertainty propagation. We have added the following sentence to the discussion: "In this paper we have concentrated on the major sources of discrepancy between the different instrumentation (such as motion and FOV) and correcting the data for these effects using statistics or regressions. However, other sources of uncertainty are still inherent within the data and an analysis of e.g. temperature dependency, cosine response, response time and instrument attitude within the framework of an error propagation model (Miller et al., 2004) are required to fully understand the instrument characteristics. Uncertainties generated in the calibration procedure have been highlighted in this paper as an offset correction to the AOD calculation. It is also likely there has been long-term instrument calibration drift during the period described in this paper, which has not been accounted for in our calculations. Some of the differences shown in Figure 4 between the Mt. Teide and Valencia Langley plots (separated by a period of 1 – 2 years; see Table 3) could be due to this factor, although there is an additional complication of inter-site differences (altitude, atmospheric composition). Calibration drift may also play a part in the comparison between the PAR and Irradiant Energy observations (Figure 12) as a period of 4 years (recommended calibration interval is 2 years) separates the calibration points and the AMT24 cruise (Table 3). However, other sources of uncertainty exist in this case when comparing broadband with hyperspectral instrumentation, such as integration range, sensitivity and spectral response functions. The comparisons shown in Figure 12 were intended to show that the hyperspectral instruments were capable of providing realistic retrievals of broadband quantities, useful for marine environmental research."

**(R3.3)** Table 3 has been updated with details of the PAR, SPN1 and shortwave instrument characteristics as well as the HyperSAS. This now includes measurement uncertainties, FOV, stray light, spectral (sampling (pixel) resolution, spectral resolution, spectral accuracy) and temperature dependency as appropriate and applicable. The text has been updated when describing the two spectrometers to give these key aspects (see **R3.7** and **R3.8**).

**(R3.4)** Guzzi et al. reference added

**(R3.5)** Added: "Assuming a clear sky, and ideal cosine responsivity of the instrument input optics, the aerosol optical depth, $\tau_a$, can then be calculated."

**(R3.6)** Added: "m is the atmospheric air-mass, in this paper defined as:"

**(R3.7)** Added **details** to text: " … pixel resolution of around 6nm across the range 350nm – 1050nm**, at a spectral resolution (Δλ FWHM) of 13nm, with <0.2% stray light.**"

**(R3.8)** Added **details** to text: "… single Zeiss MMS1 spectrometer. This has a 256 pixel detector, giving a pixel resolution of around 3.5nm across the range 350nm – 1050nm, **at a spectral resolution (Δλ FWHM) of 10nm**. The advantage of this configuration is that the Zeiss is a very stable spectrometer **(0.3nm accuracy)** over a wide range of temperatures **(<0.01nm K$^{-1}$)**, with a high sensitivity **(10$^3$ Vs J$^{-1}$) and low stray light characteristics (<0.8%)**."

There are no details about long-term stability of the spectrometer, and this is likely to vary in the field in any case. The period over which the spectrometer was in the field was too short for us to really test this, although we have alluded to calibration drift in the discussion regarding figure 4.

**(R3.9)** Added details to the text: " … the spectrometers were first calibrated using a 300mm diameter integrating sphere illuminated by a halogen lamp to give a uniform diffuse irradiance across all the 7 sensors. The irradiance at the integrating sphere port was calibrated to an Ocean Optics LS-1 calibration lamp to give an approximately correct calibration for each sensor. In particular, because the halogen lamp has a smoothly varying spectral distribution, the relative values will be correct over moderate wavelength intervals, even if the absolute scaling is incorrect. Following this, the spectral calibration was adjusted using the Langley method on Mt Teide, Tenerife (2300m, near the base of the teleferico). The calibration was adjusted smoothly across the whole spectrum using the Langley values outside the gas absorption bands, to give a final absolute calibration. The instrument outputs were calibrated to radiometric units, so the Langley calculated TOA values should match the SMARTS2 extra-terrestrial spectrum outside the of gas absorption bands."

**(R3.10)** A further explanation is added: "Figure 4 shows how these different methods compare, by plotting the extra-terrestrial irradiance values they predict. It is evident that the Langley plot performed at Mt Teide closely matches the SMARTS2 spectrum due to the site pristine conditions, except for the gas absorption bands where the Langley method cannot be applied correctly. The effect of the gases in these bands is even clearer for the Langley extra-terrestrial spectrum obtained at Valencia, as the water vapour amount is higher at sea level. In any case the absorption bands will not be used for deriving the aerosol optical depth."

**(R3.11)** As we have stated in the discussion, this paper is intended to describe the general construction and potential application of the hyperspectral radiometers. We have noted areas for future work in calibration and analysis, and this would include a more rigorous analysis of accuracy and uncertainty.

Within the AOD calculation chain, the SMARTS2 extra-terrestrial spectrum is always used as reference. This has been clarified in section 2.1: "After calibration, the spectrometer system gives outputs in radiometric units, so the top of atmosphere values give an extra-terrestrial spectrum which should match the SMARTS2 model. The SMARTS2 spectrum is used as reference in subsequent AOD calculations."

**(R3.12)** The section no longer exists in this form.

**(R3.13)** Re-reading the sentence showed that it could have been ambiguously interpreted. The sentence has been reworded: **"As a direct consequence of this, the subsequently calculated AOD values show less variability during stable periods."**

In other words, the data have been corrected for motion (giving less variability in the data) and therefore the AOD values, calculated from these corrected values, give correspondingly less variability.

**(R3.14)** Table 3 has been updated to contain this information: the spectral resolution of the HyperSAS instrument is 10nm, and has a sampling resolution of ~3nm. The estimated uncertainties on the measured HyperSAS irradiances are $10^{-5}$ Wm$^{-2}$nm$^{-1}$ (Noise Equivalent Irradiance) with another 3% on the cosine response for close to zenith. Added the sentence: **"Visually from Figure 13, the largest inter-sensor differences occur within absorption bands, therefore different spectral resolutions may play a role in these regions."**

**(R3.15)** Figure 10 (now Figure 6) is likely to contain points which have not been perfectly screened for clouds (by either instrument). However, these points are few and far between as the density plot shows that there are many more points which tend towards the 1:1 relationship. Added the sentence: **"Some of the outliers shown in Figure 6 are likely to be caused by imperfect cloud screening of data from either or both sensors"**. The data from 1020nm have been excluded in this analysis as in general we were concerned about data from this wavelength (as it is close to the operational range of the detector (1050 nm), and sensitivity is very low) and the need for an increasing correction due to temperature (the POM detector is temperature stabilised). Concentrating on 400, 500, 675 and 870 nm also gave consistency with the previous analyses for the other instrumentation.

**(R3.16)** Much of the behaviour in Figure 11 (now Figure 7) is subsequently discussed further on in the text concerning the correction algorithm for the FOV (modelling and observations). The reason for the seeming non-linearity could be as a function of airmass. However, there could be additional factors which we have not yet discovered.

**(R3.17)** We have rerun the simulations using the SMARTS2 model and found that the results were highly dependent upon the determined value of the extra-terrestrial irradiance (perhaps unsurprisingly). We have updated the text as follows and have a new version of figure 8 (was figure 12 in the previous version of the text).

"The difference between shadowband radiometer and sun photometric retrievals of AOD has previously been observed, and subsequently empirically corrected for by di Sarra et al. (2015), and attributed to the radiant impact of aerosol forward scattering on different instrumental FOV. Here we investigate this further with a modelling study using the SMARTS2 (Gueymard, 2001) solar model. This has the facility for calculating the spectral $I_N$ received for different aerosol conditions and different detector FOVs. The model was run for a range of different solar zenith angles (0 – 85 with 10° increments) and AODs (0.01 – 0.50 in 0.01 increments), and the $I_N$ calculated for a detector FOV of 7.5°, at 500nm. The AOD that would be calculated from the measured $I_N$ using the spectrometer AOD equations 1 – 8 was compared with the AOD value input into the model (Figure 8). This shows two distinct features that aid in interpreting the intercomparison with the CIMEL (Figure 7): (1) a regression slope of approximately 0.8; (2) a slight dependency on solar zenith angle. Significantly however, the over prediction of AOD at low atmospheric turbidities (AOD < 0.1) is not reproduced. This behaviour can be replicated by introducing small calibration errors to the model data. At 500nm the extra-terrestrial irradiance used in the SMARTS2 model is 1.932Wm$^{-2}$nm$^{-1}$, but in the region between 495nm and 505nm (typical instrument bandwidth of 10nm) it varies between 2.059 (497nm) and 1.878 (502nm). This range of values can account for a variation in the retrieved AOD$_{500}$ of

approximately 0.08. Not only does this highlight the importance of the accuracy of the instrument calibration, but also an understanding of instrument characteristics are required (spectral response function and resolution). At low optical depths, even in the most transparent atmospheric window, gaseous absorption is also likely to play a role in accurately determining AOD."

5 **(R3.18)** Added an insert map to figure 14 showing the position of the Microtops measurements, and caption updated to read: "…Map of the Microtops sampling locations shown in upper left of figure." Clarified the frequency of measurements by adding the following "Figure 14 shows these results plotted against latitude for the entire cruise for both Spectrometer 2 (Zeiss) and the Microtops **(observations shown are single retrievals measured daily around solar noon if conditions allowed)**."

10 **(R3.19)** Updated the end of the paragraph on the intercomparison with the Microtops to read: "However, the previous studies alluded to above have been for land based observations and therefore no uncertainties due to the platform motion are present. Moreover, the Microtops AOD has a somewhat larger uncertainty than the CIMEL."

**(R3.20)** The results presented in this paper were exclusively for Spectrometer 2 and not Spectrometer 1. We have updated the paragraph to read: "
[revised manuscript text omitted]